# The Utilization of 15-Minute Multidisciplinary Rounds to Reduce Length of Stay in Patients under Observation Status

**DOI:** 10.3390/healthcare11212823

**Published:** 2023-10-25

**Authors:** Swapnil V. Patel, Abbas Alshami, Steven Douedi, Ndausung Udongwo, Mohammad Hossain, Dana Tarina, Brian Walch, Kim Carpenter, David Kountz, Vito Buccellato, Kenneth Sable, Elliot Frank, Arif Asif

**Affiliations:** Department of Medicine, Jersey Shore University Medical Center, Hackensack Meridian Health, Neptune, NJ 07753, USA; swapnil.patel@hmhn.org (S.V.P.); abbas.alshami@hmhn.org (A.A.); douedis@deborah.org (S.D.); mohammad.hossain@hmhn.org (M.H.); dana.tarina@hmhn.org (D.T.); brian.walch@hmhn.org (B.W.); kim.carpenter@hmhn.org (K.C.); david.kountz@hmhn.org (D.K.); vito.buccellato@hmhn.org (V.B.); kenneth.sable@hmhn.org (K.S.); elliot.frank@hmhn.org (E.F.); arif.asif@hmhn.org (A.A.)

**Keywords:** discharge planning, length of stay, quality improvement

## Abstract

With the recent change to value-based care, institutions have struggled with the appropriate management of patients under observation. Observation status can have a huge impact on hospital and patient expenses. Institutions have implemented specialized observation units to provide better care for these patients. Starting in January 2020, coinciding with the initiation of daily multidisciplinary rounds, our study focused on patients aged 18 and older admitted to our hospital under observation status. Efforts were built upon prior initiatives at Jersey Shore University Medical Center (JSUMC) to optimize patient care and length of stay (LOS) reduction. The central intervention revolved around the establishment of daily “Observation Huddles”—succinct rounds led by hospital leaders to harmonize care for patients under observation. The primary aim was to assess the impact of daily multidisciplinary rounds (MDR) on LOS, while our secondary aim involved identifying specific barriers and interventions that contributed to the observed reduction. Our study revealed a 9-h reduction in observation time, resulting in approximately USD 187.50 saved per patient. When accounting for the period spanning 2020 to 2022, potential savings totaled USD 828,187.50 in 2020, USD 1,046,062.50 in 2021, and USD 1,243,562.50 in 2022. MDR for observation patients led to a reduction in LOS from 29 h to 20 h (*p* < 0.001).

## 1. Introduction

Observation care, as defined by the Centers for Medicare and Medicaid Services (CMS), encompasses specific, clinically appropriate services that involve short-term treatment, assessment, and reassessment to determine the need for further hospital inpatient treatment or discharge [1,2,3,4,5]. Typically, patients placed under observation status stay for approximately 24 to 48 h [6,7,8]. The admitting provider must subsequently decide whether to admit the patient to inpatient service or discharge them [6]. However, observation status poses challenges for patients, providers, and institutions. Being classified as outpatient care, hospital expenses for observation patients are covered by Medicare Part B, resulting in 20% cost-sharing for patients [6]. Consequently, patients often face high bills due to Medicare Part B deductibles, copays, and the absence of inpatient pharmacy coverage. Additionally, it may lead to low reimbursement from insurance companies, resulting in financial losses for healthcare institutions [9].

The increase in observation status is primarily attributed to Medicare’s Recovery Audit Contractor (RAC) program and penalties associated with readmissions [6]. The RAC program was implemented to recover unnecessary costs by reviewing the appropriateness of admissions and potentially denying full payment for inappropriate status designations [6]. Consequently, hospitals have become more cautious and started admitting more patients under observation status [6].

To address these challenges, hospitals have pursued innovative strategies. Some institutions have introduced dedicated units and standardized protocols to streamline care and reduce variability [10,11]. Amid these efforts, Jersey Shore University Medical Center (JSUMC), a leading tertiary care center, embarked on a multidisciplinary approach to alleviate the impact of expansion on patient LOS [12]. A pivotal step in this endeavor was the establishment of a dedicated observation unit in 2018, a proactive response to the evolving healthcare landscape. Building upon this foundation, JSUMC introduced daily multidisciplinary team rounds as a means to further reduce LOS for observation patients [12]. Prior to the intervention, there were no multidisciplinary rounds at JSUMC. All patients were admitted to an observation unit. This initiative aligns with studies such as those by Meo et al., who introduced electronic tracking tools within multidisciplinary discharge rounds to decrease LOS and enhance patient outcomes [13], and by Bellolio et al., who examined the impact of patient disposition on outcomes for those under observation status [8]. After the implementation of these rounds, the LOS for observation patients decreased.

Within this framework, our study aims to comprehensively analyze the impact of daily multidisciplinary rounds on patients’ LOS in the observation units. By evaluating the effectiveness of these rounds and their contribution to reducing LOS, we seek to provide valuable insights into optimizing patient care strategies and addressing the challenges associated with observation status.

## 2. Materials and Methods

### 2.1. Overview of the Project

Patients aged ≥18 who were admitted to our hospital with an “observation status” were included in this retrospective study. Data collection began in January 2020, coinciding with the introduction of daily multidisciplinary rounds. Prior to this, JSUMC made various multidisciplinary efforts to reduce the overall LOS of patients described by Patel et al. [5]. In these efforts, unit-based advanced-practitioner nurses were deployed on units including the observation unit, and improved the turnaround time (TAT) of various key testing such as echocardiography (ECHO), magnetic resonance imaging (MRI), and computed tomography (CT) scans [12]. In addition to these efforts, daily multidisciplinary team rounds were implemented to reduce LOS on observation patients specifically. Patients were discussed in order of LOS, with higher LOS patients discussed first to emphasize that they needed faster work up and disposition.

### 2.2. Context

JSUMC is a prominent tertiary teaching medical center located in central New Jersey. It encompasses a total of 646 beds and offers seven residency and six fellowship programs. Among the available beds, excluding those dedicated to psychiatry and women’s and children’s services, there are 413 beds. This includes 62 critical care beds, 81 beds distributed across seven medicine/surgery units, and 270 beds spread across seven telemetry units.

### 2.3. Quality Improvement Process

To address the escalating patient caseload and LOS, a collaborative and interdisciplinary team was established. Composed of professionals from diverse departments, including medical staff, medical residency program, nursing leadership, front-line nursing, process improvement, environmental services, and operations, the interdisciplinary team was established with the backing of senior leadership. Our study focuses on elucidating the comprehensive multidisciplinary approach employed to identify barriers and develop a multifaceted strategy that resulted in a reduction in LOS. Particularly for observation patients who require time-sensitive care, a dedicated multidisciplinary team was assembled to streamline their treatment and optimize outcomes.

### 2.4. Interventions

The primary intervention involved the implementation of an “Observation Huddle”, a daily 15 min multidisciplinary briefing designed to coordinate the care of patients admitted under observation status. Starting in January 2020, these rounds occurred every day of the week and took place from 9:15 a.m. to 9:30 a.m. The observation round team was composed of individuals outlined in Table 1 with guidance provided by a designated physician leader. Each participating team member was equipped with a comprehensive list of patients receiving observation care. For each patient, the attending physician presented a brief overview of the case, including the current patient status and highlighted any existing patient-specific obstacles. For instance, a presentation might be “Mr. Jones in room B504 presented with chest pain and is awaiting an exercise stress test. Also, transportation home is unavailable”. Subsequently, team members responsible for pending patient matters (e.g., the cardiac diagnostic testing manager and the case management team in the case of Mr. Jones) indicated the outstanding tasks. Consequently, at the conclusion of the huddle, every team member had a checklist detailing the necessary steps to expedite observation patient’s care. Following the huddle, team members assumed the responsibility of updating the attending physician on the progress of each pending task. In turn, the attending physician conveyed these updates to the physician leader. This approach allowed the physician leader to maintain a current and comprehensive overview of observation patient statuses, thereby facilitating the accurate identification of systemic barriers that impede patient care optimization. Furthermore, the unit-based advanced practitioner nurse (APN) forwarded an end-of-day report to team members, reinforcing coordination efforts and offering a platform to address any persistent barriers as required.

### 2.5. Primary Outcome

The primary outcome that was measured was the total hours a patient remained on observation status at our medical center.

### 2.6. Secondary Outcomes

The secondary outcome was the measurement of cost for patients admitted under observation status.

### 2.7. Funding and Ethical Consideration

This dataset underwent scrutiny within the context of a quality improvement process. Given that we collectively assessed the data and did not individually review patient cases, the requirement for institutional review board approval was not applicable. Furthermore, this research did not receive dedicated funding from any public, commercial, or not-for-profit organization.

## 3. Results

### 3.1. Length of Stay

Prior to the intervention, the mean number of visits was 384 visits per month, while the post-intervention number of visits was 459 visits, and the *p*-value was 0.03. The average pre-intervention was 29.11 h (SD 1.78), while the post-intervention average was 20.14 h (SD 3.03), *p*-value < 0.001. The above was using an independent sample Student’s *t*-test. The mean hours per month are shown in Figure 1.

### 3.2. Financial Impact

From prior analysis, we estimate that the average cost per day to treat a patient is USD 500 [5]. With a reduction of 9 h of observation time, an estimate of USD 187.50 per patient is saved. With consideration volume from 2020–2023, there were potential cost savings of USD 828,187.50 in 2020, USD 1,046,062.50 in 2021, and USD 1,243,562.50 in 2022. This amounts to total cost savings of USD 3,116,812.50 over 3 years. With 6627 observation stays and a 9 h reduction in LOS, it is estimated that approximately 6.8 beds per day were created.

## 4. Discussion

Our study highlights the significant reduction in LOS achieved by implementing a daily observation huddle for patients under observation status. The utilization of the observation huddle not only reduces LOS but also yields substantial financial benefits for patients and institutions [14].

In prior efforts, various institutions have implemented a specialized observation unit to provide optimal care for patients under observation. These units required strong leadership, coordinated care, and careful monitoring for patient safety and quality purposes [15]. Prior studies have shown that a dedicated unit can create an impact on the reduction in LOS observation patients [16,17]. At our institution, a clinical decision or observation unit was established well prior to our intervention. Similar to prior studies, our institution’s clinical decision unit was staffed with a multidisciplinary team. The implementation of a multidisciplinary round for the unit made a significant decrease in observation hours.

Our study’s results demonstrated a significant reduction in the LOS among observation patients through the implementation of daily multidisciplinary rounds. The pre-intervention mean LOS of 29.11 h was substantially decreased to an average of 20.14 h post-intervention (*p* < 0.001). This reduction highlights the efficacy of our intervention in streamlining patient care processes and expediting the decision making for patient disposition. Our findings are consistent with the outcomes of similar studies, such as the work by Nimmagadda et. al., who utilized virtual multidisciplinary rounds to reduce LOS and promote accountability [18].

Their approach, albeit different in nature, underscores the importance of coordinated care in optimizing LOS outcomes. The financial impact analysis further underscored the benefits of our intervention. With an estimated average cost of USD 500 per patient day, the reduction of 9 h of observation time translated to approximately USD 187.50 per patient saved. Notably, when considering patient volume from 2020 to 2022, our study projected potential cost savings of USD 3116, 8212.50 over three years. These findings align with Dada and Aule’s study, which identified factors influencing LOS for observation patients [19]. Our work complements their findings by offering a tangible financial perspective, elucidation how even modest reductions in LOS can yield substantial cost savings and resource optimization.

The utilization of multidisciplinary rounds and team effort have been shown to affect various quality improvement measures, such as a reduction in LOS, a reduction in telemetry utilization, and the improvement of coordination of care. Prior studies have shown that utilization of multidisciplinary rounds created a positive impact on early discharges, LOS, as well as a reduction in readmissions and improved patient satisfaction [20,21,22,23]. Similar to prior studies, we have shown that the daily multidisciplinary coordination of care can lead to a reduction in LOS. We described various barriers in our prior study to the reduction in LOS, which included timely completion of diagnostic testing such as MRI, ECHO, and CT scans; a lack of early ambulation; and prolonged LOS with private providers [12]. The coordination of care with these services in the multidisciplinary rounds played a significant role in the reduction in LOS.

Many healthcare systems faced challenges of capacity management and throughput. Due to high volumes, institutions are forced to expand hospital infrastructure, which can lead to high costs. This intervention highlights the creation of beds with the optimization of operational procedures. Further financial impact can be implied from the potential cost savings needed to create more physical space, as well as the revenue that is generated with the creation of the available beds [24].

While our study provides valuable insights into the impact of daily multidisciplinary rounds on reducing LOS in observation patients, several limitations should be acknowledged. Firstly, the study was conducted in a single medical center, JSUMC, which may limit the generalizability of our findings to other healthcare settings with different patient demographics and resource allocations. Additionally, the study’s retrospective nature could introduce selection bias and potential confounding factors that were not accounted for. The financial impact calculations were based on estimates and assumptions regarding patient costs, and variations in healthcare pricing could affect the accuracy of the projected savings. Throughout the study duration, our institution maintained a consistent level of disease severity and case mix. It is crucial to recognize, however, that our analysis did not encompass an exhaustive evaluation of individual patient characteristics or variables. This omission imposes limitations on our capacity to comprehensively account for potential confounding factors that might influence the observed outcomes. Furthermore, the effectiveness of the “Observation Huddle” intervention may depend on the availability and expertise of the multidisciplinary team members, which could vary across institutions. Lastly, while the reduction in LOS is a promising outcome, the other important metrics, such as patient satisfaction and readmission rates, were not extensively explored in this study. Future research should aim to address these limitations and conduct multi-center studies to validate our findings in diverse healthcare settings.

## 5. Conclusions

The implementation of daily multidisciplinary rounds with key clinical staff can lead to a reduction in LOS in patients under observation. Institutions should not only create a specialized unit but also coordinate aspects of care that can lengthen observation stay, in order to reduce observation time and improve financial gain.

## Figures and Tables

**Figure 1 healthcare-11-02823-f001:**
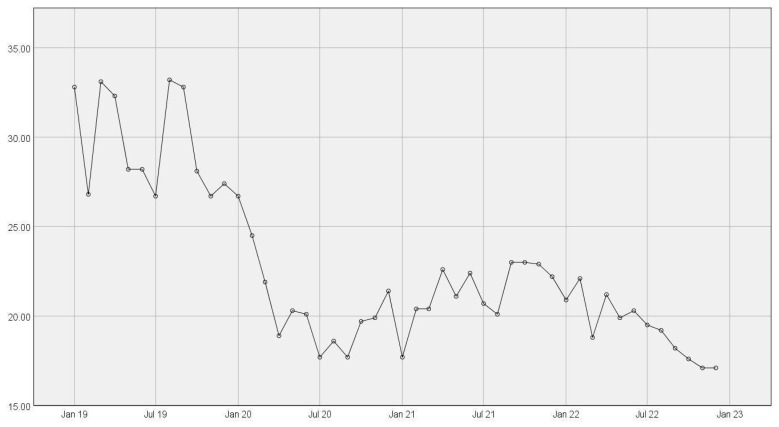
Mean hours spent on observation by month—this figure outlines the average hours a patient spent in observation per month. Multidisciplinary rounds were implemented in January 2020.

**Table 1 healthcare-11-02823-t001:** Team members participating in daily observation huddles.

Team Leaders	Designated Task
Physician Leader	Coordination of rounds by leading patient case discussion and care plan
Attending Physician(s)	Primary attendings on observation patients Present care plans at the huddle to facilitate coordination
Nursing Manager	Represent nursing staff to address and coordinate nursing care plan
Case Management	Review disposition planning and coordinate discharge needs
Unit-Based Assistant Nursing Practitioner	Assist physicians with care plan
Physician Advisor	Review patient insurance status, and possibility for inpatient admission criteria
Cardiac Diagnostic Testing Manager	Facilitate expedited stress testing and echocardiography
Radiology Manager	Facilitate expedited imaging for observation patients
Vice President of Clinical Operations	Executive sponsor for rounds

## Data Availability

All data used for this study are available within the manuscript.

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
