# Peer review of "The Utilization of 15-Minute Multidisciplinary Rounds to Reduce Length of Stay in Patients under Observation Status"

_healthcare, 2023, doi:10.3390/healthcare11212823_

Round 1

Reviewer 1 Report

Comments and Suggestions for Authors

Dear Respectable Authors

Thank you for considering a significant area of research related to LOS. Your research is of interest to the reader but needs some revisions as follows;

- The title is not clear enough. "Patients under observation status" or "at observation unit" may be better. 

- Add a brief of methods in the abstract section. 

- Add more results in the abstract section, especially regarding cost.

- Please add your study aims at the end of the introduction section.

- Please add limitations of your study.

- Add some statements regarding funding and ethical consideration.

- The number of references is not enough. 

- The way of writing abbreviations is not good. It should be equalized. For some words, abbreviations have been repeated several times. Some words are listed only in their abbreviated form. Please use the full and abbreviated form the first time and only the abbreviated form in the rest of the text (highlighted in the text). 

- Table 1 needs to be restructured. It is better to list the names one after the other.

- discussion section is low. Please interpret all your results in the discussion section. Please add a discussion regarding cost. the number of references used in the discussion section is not quite enough.

Cheers

Comments on the Quality of English Language

None

Author Response

Good morning, 

Thank you so much for taking time out of your busy schedule to review our manuscript. We ensured that all your suggestions for the paper were made. 

  1. Title: The title was changed based on what you suggested
  2. Abstract: A Brief method section was added to the abstract as well as the results, including the cost 
  3. Abbreviations: We went through the abbreviations from top to bottom to ensure they were in order (JSUMC, LOS, ECHO, CT, MRI, etc)
  4. Ethics & Funding: We added the ethics and funding statement in the method section as advised. 
  5. Table 1: This Table has been restructured as advised
  6. Introduction: We added the aim of the study paper to the introduction. 
  7. Discussions: We added about 3 more paragraphs to the discussion 
  8. Limitations: We include the limitations of the study
  9. References: We added More references (13 more references) to the paper as advised 

In summary, we made changes to what you suggested. Thank you again for helping us improve our paper. 

Regards, 

Swapnil Patel, MD. 

Deputy Chief Internal Medicine, 

Jersey Shore University Medical Center. 

Reviewer 2 Report

Comments and Suggestions for Authors

This is an interesting topic, and an observation method was used. The article is organized and well-written. However, I was unable to understand,  what was purpose of the multidisciplinary rounds?. Whether the rounds were designed for research purpose or as part of the hospital procedure. Is there any prior research? If so, it should be included in the introduction, along with supporting evidence. Furthermore, there are numerous overlaps and methodological concerns in the methodologies section, such as data analysis, patient characteristics (Variables), and so on. The entire manuscript must be thoroughly revised.

Author Response

Good morning, 

Thank you so much for taking time out of your busy schedule to review our manuscript. Here are the corrections we made to your suggestions 

  1. We added the aim of our study to the paper in the introduction, stating why we made the multidisciplinary rounds. This was tailored to reduce the length of stay in Obs patients. 
  2. We included prior research in the introduction as advised and also added more references and limitations to the study. 
  3. Our limitation also included patient characteristics (variables) which we included as part of our limitation in the final paragraph of the discussion. 

Thank you. 

Regards, 

Swapnil Patel, MD. 

Deputy Chief Internal Medicine, Jersey Shore University Medical Center.

Reviewer 3 Report

Comments and Suggestions for Authors

The article submitted for review is very interesting and deals with an important issue concerning coordinated hospital care. On one hand, it concerns the aspect of the length of a patient's stay in a hospital ward, which also has a direct impact on the economic aspects of the healthcare system. On the other, the same length of stay can have an impact on the psychological feelings of the patient as well as improving the diagnostic process. The proposed intervention is very interesting, especially because of the creation of a multiprofessional treatment team. 

A weak part of the article is the poor characterisation of the study group. Were patients before and during the intervention similar in terms of, for example, disease entities, reported symptoms, diagnostic difficulties? Such information should be included in the text, as it can have a significant impact on the interpretation of the results regarding the length of stay of the patient.

Author Response

Good morning, 

Thank you so much for taking time out of your busy schedule to review our manuscript. Also, thank you for your kind words. 

  1. We agree with your assessment and comments for the paper, hence, we included the limitation in patients' characteristics (variables) in the last paragraph of the discussion section which was not present in the first draft. 

Thank you. 

Regards, 

Swapnil Patel, MD. 

Deputy Chief Internal Medicine, Jersey Shore University Medical Center. 

Reviewer 4 Report

Comments and Suggestions for Authors

Clearly present the research method, clearly present the concept of “15-Minute Multidisciplinary Round”, and what characteristics the number of participants in the study had. In addition, it would be appreciated if you clearly indicate when and how much the intervention was implemented and submit it again.

1. Before using JSUMC on page 2, indicate Jersey Shore University Medical Center (JSUMC) on page 1 before using JSUMC.

2. Present a conceptual framework for your research in figure 1.

3. The team members are indicated in Table 1. Are they composed of one person each? Also, describe what qualifications they have.

4. Please clearly present the work you do during the 15-minute round with team members in Table 1.

5. Please clearly indicate where the pre-intervention ends and where the post-intervention starts in Figure 1.

6. In the research method, clearly state when and to whom the intervention was provided.

7. No description of research ethics considerations. This needs to be supplemented.

Author Response

Good morning, 

Thank you so much for taking time out of your busy schedule to review our manuscript. We agree with your comments and suggestions, hence, there are several things we added/changed based on what you advised. 

  1. 1. Before using JSUMC on page 2, indicate Jersey Shore University Medical Center (JSUMC) on page 1 before using JSUMC.

    Thank you. This has been added one page 2 as well as the abstract.  

    2. Present a conceptual framework for your research in figure 1.

     Thank you. This was added in the description of figure 1, as well as in the text. 

    3. The team members are indicated in Table 1. Are they composed of one person each? Also, describe what qualifications they have.

     Thank you. There was an additional column that was added that detailed each team members roles. 

    4. Please clearly present the work you do during the 15-minute round with team members in Table 1.

     Thank you. This was added to the methods portion in which there was details about the work done on rounds. 

    5. Please clearly indicate where the pre-intervention ends and where the post-intervention starts in Figure 1.

     Thank you. This was added to Figure 1. 

    6. In the research method, clearly state when and to whom the intervention was provided.

    Thank you. This was added in the "intervention" part in the methods sections.  

    7. No description of research ethics considerations. This needs to be supplemented.

    Thank you. This was added.    At the same time, we noticed that the number of words in manuscript is 2388,
    which is still less than 4000 words. If it is convenient, could you please
    provide the reason why it cannot be expanded?
      Respected Editors and Reviewers,   We thank you for the review of our manuscript. We have acknowledged your comments on the total words of the manuscript. We believe that there is limited meaningful or relevant literature for management of observation patients. We believe our manuscript is detailed yet concise, which will make it easier for readers to understand and potentially implement a solution similar to ours at their institution. We believe that adding additional words may make the manuscript longer, however not significantly contribute to the quality. We are happy to however add anything the editors or reviewers still feel may be relevant or helpful. Thank you for your feedback!

Thank you. 

Regards, 

Swapnil Patel, MD. 

Deputy Chief Internal Medicine, Jersey Shore University Medical Center.

Round 2

Reviewer 2 Report

Comments and Suggestions for Authors

Dear Authors, 

Thank you for addressing all my comments. 

Author Response

You are very welcome. Thank you for giving us feedbacks

Reviewer 4 Report

Comments and Suggestions for Authors

1. Since you are examining the effectiveness of “various multidisciplinary efforts to reduce the overall LOS of patients,” it is essential to clearly state what efforts consist of. Be sure to describe the core components of “the observation unit” and “improved turnaround time (TAT)” and clearly indicate how it differs from existing care. (in terms of time, organization, treatment, etc.)

ex) The current care is fundamentally different from the existing care in some respects, oo has been newly included, xx time has been increased...

2. Are obsevation hours important in this study? Figure 1 shows the conceptual framework of the entire study.

3. In the text, M1 to M11 are indicated. What do they mean? If you wish to clarify this and include modified content, please insert them accordingly.

Author Response

Good afternoon, thank you for the feedback 

  1. Since you are examining the effectiveness of “various multidisciplinary efforts to reduce the overall LOS of patients,” it is essential to clearly state what efforts consist of. Be sure to describe the core components of “the observation unit” and “improved turnaround time (TAT)” and clearly indicate how it differs from existing care. (in terms of time, organization, treatment, etc.)ex) The current care is fundamentally different from the existing care in some respects, oo has been newly included, xx time has been increased...

answer: In our method section we were only able to report what was done during this project. 

2. Are observation hours important in this study? Figure 1 shows the conceptual framework of the entire study.

Answer: Yeah it is. 

3. In the text, M1 to M11 are indicated. What do they mean? If you wish to clarify this and include modified content, please insert them accordingly.

answer: M1 to M11 is a hidden comment when a Word document is
converted to a PDF file, we didn't include those in our word document. We added the conflict of interest statements in the revised manuscript.